# Relationship between Seed Morphological Traits and Ash and Mineral Distribution along the Kernel Using Debranning in Durum Wheats from Different Geographic Sites

**DOI:** 10.3390/foods9111523

**Published:** 2020-10-23

**Authors:** Donatella B.M. Ficco, Romina Beleggia, Ivano Pecorella, Valentina Giovanniello, Alfonso S. Frenda, Pasquale De Vita

**Affiliations:** 1Consiglio per la Ricerca in Agricoltura e l’Analisi dell’Economia Agrarian—Centro di Ricerca Cerealicoltura e Colture Industriali, S.S. 673 km 25.200, 71122 Foggia, Italy; romina.beleggia@crea.gov.it (R.B.); ivano.pecorella@crea.gov.it (I.P.); valentina.giovanniello@crea.gov.it (V.G.); pasquale.devita@crea.gov.it (P.D.V.); 2Dipartimento Scienze Agrarie, Alimentari e Forestali, Università degli Studi di Palermo, Viale delle Scienze, 90128 Palermo, Italy; alfonso.frenda@unipa.it

**Keywords:** durum wheat, kernel shape and size, debranning, ash content, macro- and micro-elements

## Abstract

Debranning was applied to durum wheat to the study the relationship between kernel shape and size, and ash and mineral distribution having implications for semolina yield. To this aim four durum wheat genotypes carried out over three environments were selected to determine the morphological and yield traits as well as the distribution along the kernel of the ash, macro- (Na, K, P, Ca, and Mg), and micro-elements (Mn, Fe, Cu, Zn, and Mo). A descendent ash gradient within the kernel reflects the decreases in the minerals that occurred during debranning. Perciasacchi with high seed weight (TKW) and greater thickness followed by Cappelli showed a more uniform distribution of ash content along the kernels. High *r* Pearson coefficient (*p* < 0.01) showed an inverse relationship between thickness and ash decay. Since thickness was strongly affected by the genotype, it could represent a useful trait for breeding programs to predict the milling quality.

## 1. Introduction

Durum wheat (*Triticum durum* Desf.) is the most cultivated cereal in the Mediterranean area and consumed for its products as semolina, cous cous, bulgur, pasta, and bread.

Breeding durum wheat activity over the past century has devoted to release of high-yielding cultivars with improved end-use quality (protein concentration, gluten strength, and semolina pigment content) [1,2,3].

The semolina yield is another important quality factor for the economics of milling industry, unfortunately its determination is laborious and requires the use of specialized equipment that cannot be easily integrated into a breeding program. For this reason, the breeding programs for this trait were limited whereas studies were carried out indirectly through image analysis of secondary traits such as grain morphology or other attributes related to milling quality [4,5,6,7,8].

The relationships between grain size and shape and milling yield were investigated by several authors [9,10,11]. These studies developed multiple regression models based on kernel characteristics (i.e., length of minor and major axes, perimeter, area, and ellipsoidal volume) for the prediction of milling quality of durum wheat without conducting actual milling. In a recent study performed by Haraszi et al. [12], a model based on results of the single-kernel characterization system accounted for 65% of the variance in total milling yield of durum wheat. Unfortunately, the kernel traits used by the model could be susceptible to environmental variation and further validation with wheat samples grown at different locations and in different crop years is necessary. In addition, the factors that can influence the semolina yield include both physical (e.g., vitreousness, hardness, and test weight) [13,14] and chemical characteristics (i.e., ash content) [15].

Milling industries in several European countries consider ash content as the most important durum wheat characteristic. For example, Italian law (DPR 187/2001) imposes limits on ash content in durum wheat products used for human food (i.e., 0.9% dmb for semolina and pasta). Semolina ash content is correlated both with ash content of the whole kernel [16] and with the extraction rate [17].

The ash in wheat is not evenly distributed throughout the kernel [18], increasing from the center to the outer layers of the durum wheat grain, and affecting the milling extraction rate because of a more difficult separation of endosperm from the bran layers [19].

Debranning or pearling is a useful process based on the sequential removal by abrasion of the outer grain layers, from the pericarp up to the aleurone layers, which, applied prior to milling, showed positive effects on semolina yield, color, and contaminants [18,20,21]. Additionally, semolina color has been linked to ash content to assess the cleanliness of semolina.

Studies have proved that ash content is influenced by genotype-environment interaction [13,15,18,22,23], so the selection of genotypes with a more favorable ash distribution in the external parts of the grain could improve the potential of semolina yield. To this end, kernel size, shape and weight could also have an effect on ash distribution in wheat kernel as demonstrated previously by Troccoli et al. [15] and Fares et al. [18]. Kernel size and shape are related to milling potential, to reach the best separation of kernel parts during processing. Baasandorj et al. [24] found that small kernels had low semolina yield due to proportion of starchy endosperm. Troccoli and Di Fonzo [8] found a positive correlation between test weight and flour yield. Small-sized kernels had the lowest test weight and semolina yield because of the low percentage of endosperm relative to bran and aleurone. The same conclusion was reached by Posner and Hibbs [25] who reported that large kernels with a higher percentage of endosperm showed higher semolina yield than smaller ones.

In the present work, we investigated the relationship between the morphological grain characteristics, captured by using a high-throughput image analyzer, the yield-traits and the ash content and macro- and micro-elements concentration in durum wheat. Exploiting the debranning process, we evaluated the ash distribution in the kernels of four durum wheat varieties grown in three areas of Southern Italy. The innovative approach is based on the combination of more traits, from seed morphology, yield to ash and minerals, in order to evaluate their variations in relation to different genotypes and environments.

## 2. Materials and Methods

### 2.1. Plant Materials and Field Trials

Four old tetraploid wheat genotypes (*T. turgidum* ssp.) were chosen to conduct this study, three belonging to the durum subspecies (Russello, Timilia, and Cappelli) and one to the turanicum subspecies (Perciasacchi) with different morphological grain traits. The genetic materials were grown in the 2013/2014 growing season at the farms of CREA-CI in Foggia, Italy (41°28′ N, 15°32′ E; 75 m a.s.l.) on a clay-loam soil (Typic Chromoxerert), at Pietranera farm in Agrigento, Sicily, Italy (37°30′ N, 13°31′ E; 178 m a.s.l.), on a deep, well-structured soil classified as a Chromic Haploxerert (Vertisol) and at Alimena farm in Palermo, Italy (37°74′ N, 14°12′ E, 740 m a.s.l.) on a sandy-clay soil using a randomized complete block design, with three replicates. Plots consisted of 8 rows, 7.5 m long, corresponding to an area of about 10 m^2^. The sowing dates were on December 4th, 11th, and 15th 2013 at Foggia, Agrigento, and Palermo, respectively. In the three localities a common agronomic management was adopted, using a seedling density of 350 seeds m^−2^ and splitting the fertilization into two applications one-third before sowing as ammonium phosphate, and two-thirds N top-dressed applied at the beginning of wheat tillering, corresponding to Stage 21 of the Zadoks scale (Zadoks, 1974) as ammonium nitrate. Weeds within the growing season were controlled by means of specific herbicides: Tralcossidim (1.7 L ha^−1^) + Clopiralid + MCPA + Fluroxypyr (2.0–2.5 L ha^−1^). The collection was carried out with a combine on June 21st, 12th, and 16th, 2014, respectively in Foggia, Agrigento, and Palermo. Cleaned grain samples were stored at 4 °C until analysis.

### 2.2. Sample Debranning and Milling

Approximately 1000 g of each durum wheat grain samples was cleaned, selected, and then divided into eight 100-g subsamples; one was milled to wholemeal, one was used to produce semolina and the remaining six were debranned by using a friction debranning machine (Satake, Toshiba Corp., Tokyo, Japan). Following the method of Borrelli et al. [20], recently reported in Ficco et al. [21], six sub-samples of each durum wheat grain were debranned for six increasing times of 30, 60, 90, 120, 150, and 180 s. The weight of each debranned grain and the removed material at each debranning time was weighted and their percentage respect to initial weight was calculated.

The resulting debranned kernels that correspond to the debranning treatments (DK1−DK6) removing the bran layer in average the 5%, 10.4%, 15.7%, 19.7%, 23.1%, and 26.5%, with respect to the whole grain were collected. Intact grains and grains after each debranning time were milled to obtain the wholemeal flour using a laboratory mill (Tecator Cyclotec 1093; International PBI, Milan, Italy) to pass through a 0.5 mm screen. Semolina was produced by a Laboratory-scale mill 4RB (Bona, Monza, Italy), after tempering the grain to 16.5% moisture.

### 2.3. Phenotypic Evaluation

#### 2.3.1. Grain Yield and Yield Related Trait Evaluation

An image analysis of the whole grain morphological traits (i.e., length, width, thickness; mm and area; mm^2^) and thousand kernel weight (TKW; g) was calculated in according to Ficco et al. [21]. Test weight (TW) was measured on 250 g sample with a Shopper chondrometer, expressing the data in kilograms per hectoliter (kg hl^−1^) as reported in De Vita et al. [1]. Grain yield (GY; t/ha) was also evaluated.

#### 2.3.2. Ash Content and Mineral Determination on Wholemeal, Semolina, and Debranned Flours

The ash content by incineration (%, *w*/*w*, dmb) was determined by triplicate analysis according to the UNI 2171 method [26].

Macro- (Na, K, P, Ca, and Mg) and micro-elements (Mn, Fe, Cu, Zn, and Mo) were determined according to Beleggia et al. [27]. Briefly, dried flours (20 mg) were digested in microwave with 8 mL HNO_3_ (69.5%)/2 mL H_2_O_2_ (30%) and then 0.2 mL diluted to 10 mL with high purity deionized water. The determination was performed by using inductively coupled plasma mass spectrometry (Agilent 7700x; Agilent Technologies, Italy), equipped with an auto-sampler (ASX-500). The plasma power was operated at 1550 ± 50 W, and the carrier and make-up gases were typically set at 0.83 L min^−1^ and 0.17 L min^−1^, respectively. A reference material was included randomly in the analytical batches, from digestion onwards (RM 1567b; from the National Institute of Standards and Technology). The analyses were carried out in triplicate. The data were processed using the MassHunter WorkStation software (Agilent Technologies, Milan, Italy) and expressed as mg/kg on a dry matter basis.

### 2.4. Statistical Analysis

Morphological and yield parameters on grains and ash on wholemeal were subjected to two-way analysis of variance (ANOVA) to estimate the effect of genotype, environment (site), and their interactions. Ash and macro- and micro-elements were analyzed by three-way ANOVA (*p* < 0.01), also adding the treatment (debranning), and the relative interactions. Pearson correlations (*p* < 0.01) were evaluated among morphological, yield and yield-related traits, and ash. Principal component analysis (PCA) was performed on the correlation matrices, standardizing the values of each variable, and using the Varimax method to maximize the orthogonal factor rotation. JMP version 8.0 (SAS Institute Inc., Cary, NC, USA) was used for the analyses.

## 3. Results and Discussions

### 3.1. Effect of G, E and G × E on Yield, Ash Content, and Seed Morphological Traits

Genotype (G), environment (E) and their interaction (G × E) strongly affected all the parameters, except for GY (G and G × E not significant) and seed area (E not significant) (Table 1). Palermo was the most productive location (3.32 t ha^−1^) where the highest ash content (2.16%) and test weight (80.1 kg hL^−1^) values were also recorded. The pedo-climatic characteristics of this site, and in particular its altitude, make this environment particularly suitable for old varieties characterized by a long vegetative phase and short grain filling phases, reflecting the higher plant height above 100 cm and the low grain yield, as were all the varieties considered in present study. The grain characteristics of Perciasacchi (*T. turgidum* ssp. *turanicum*) were different from those of all three ssp. durum varieties, especially for TKW (>60 g) confirming what is reported in the literature for this subspecies [28]. On the contrary, the Timilia variety confirmed the lower TKW and lower mean values of all the seed morphological parameters detected than the other varieties considered in the study, as this variety is characterized by the small size of the seeds, as previously highlighted by De Santis et al. [29]; Mefleh et al. [30].

As expected, ash content was strongly affected by the E and G × E as previously evidenced by [18,31]. Interestingly, about the seed morphological parameters (length, width, thickness, and area), although the ANOVA showed a significant effect for all the factors considered, the weight of the genotype was higher than environment and their interaction. Several studies, in fact, showed a high heritability for these traits and a good response to selection [4,32]. Perciasacchi was confirmed as the variety with the largest seed area (18.8 mm^2^) associated with greater seed width and thickness. Similar behavior was also showed for Cappelli, while Timilia confirmed the lowest values for all seed morphological parameters evaluated.

### 3.2. Effect of Debranning on Ash and Mineral Content

The three-way ANOVA analysis showed a significant effect for all treatments (*p* < 0.01) on ash content and on the macro- (Na, K, P, Ca, and Mg) and micro-element (Mn, Fe, Cu, Zn, and Mo) concentrations (Appendix A). In particular, the genotype and the treatment (debranning) were significant for all the parameters analyzed while the environment (site) influenced all the parameters evaluated except for Na (Appendix A).

Regarding the interactions of first (G × E, G × T, E × T) and second degree (G × E × T), in Appendix A are reported the ANOVA results showing a significant effect of interactions for most of the elements analyzed with some exceptions. In order to simplify the analysis and discussion of our results, the individual minerals were grouped into macro- and micro-elements. Figure 1 shows the average of ash and of the sum of macro- and micro-element data for each genotype grown in the three environments following six debranning treatments (DK1–6) from wholemeal to semolina.

All varieties showed a linear decrease in ash content following debranning treatments at all three experimental sites confirming the previous results of Fares et al. [18] and Borrelli et al. [20]. The decrease was greater after the first debranning treatment (DK1) respect to wholemeal and at semolina production, in relation to DK6, with some exceptions (Figure 1a). The loss of ash content after DK1 (5.0% of the kernel weight loss) was of 17.5% and 14.6% for Perciasacchi and Cappelli when compared to Russello and Timilia (7.7% in average), respectively. This resulted in a more limited ash content reduction compared to the values of 20–30% reported by Fares et al. [18] and Borrelli et al. [20], that although used the same debranning protocol, both the genetic materials and the experimental locations were different. At the end of DK2 (10.4% of weight kernel loss) higher losses were observed for Perciasacchi and Cappelli (27% and 23.5%, in average) respect to the other varieties (15.5%, in average). Therefore, larger grain varieties as Perciasacchi and Cappelli required shorter time for debranning, showing a remarkable reduction in ash contents in the initial steps of debranning. Shetlar et al. [33] dissected the kernel weight in the 3.9% for outer pericarp, 0.9% for the inner pericarp, 0.7% for the testa, and 9.0% for the aleurone layer. So, debranning up to 5% level removed the pericarp while at 10.4% also the aleurone layers were involved.

Focusing attention to the DK6 vs. semolina, only Perciasacchi had a loss of ash content of 10% in semolina demonstrating a more regular ash gradient across the kernels (Figure 1a), while the other varieties showed an average decrease of about 35%. The higher ash content recorded in Palermo determined a different effect on varieties. Cappelli, starting from higher ash content in the wholemeal, maintained a higher level also in the semolina.

The descendent gradient observed for ash content within the kernel reflected the decreases in the macro- and micro-elements that occurred during debranning treatments as observed by Sovrani et al. [34] and Giordano and Blandino [35] (Figure 1b,c). This strong relationship was also confirmed by the high correlation values (*r* = 0.97 macro- and *r* = 0.94 micro-elements) recorded between ash and elements measured in all debranned and milled samples.

Considering the macro-elements (Figure 1b), Cappelli showed the highest wholemeal values in all the three environments (14.647 mg/kg, in average), followed by Russello (14.074 mg/kg, in average). The levels of macro-elements for all genotypes decreased progressively with increasing debranning intensity and after milling (Figure 1b; Appendix A). Similarly to ash content, Perciasacchi showed a small loss of macro-elements from DK6 to semolina (10% in average) while a more evident decrease was highlighted for Timilia, Russello and Cappelli (41.9%, 24.2%, and 19% in average, respectively).

A decreasing of micro-elements content was observed in wholemeal with the following order Cappelli > Russello > Timilia > Perciasacchi (Figure 1c). During the debranning treatments and after milling a decay of micro-elements was observed in all genotypes with different extent, in particular from DK6 to semolina, more marked for Timilia, Cappelli and Russello, (40.7%, 36.5%, and 30.2%, in average, respectively) respect to Perciasacchi (20.9%, in average). Perciasacchi, in the present study, was confirmed a genotype with the major retention capacity of micro-element in semolina (59.16 mg/kg vs. 48.07 mg/kg in average of the other genotypes over the three environments). Among the experimental sites, highest semolina values were observed in Perciasacchi at Palermo for Fe and at Agrigento and Foggia for Zn (Appendix A).

The macro- and micro-elements distribution of Perciasacchi could be associated with the morphological seed traits. The greater area and seed weight (TKW) of Perciasacchi due to the greater length, width, and thickness of the kernels respect to Russello and Timilia could affect the distribution of mineral concentration along the kernel as reported by other authors [18,19,36]. Among the durum wheat varieties, only Cappelli showed morphological seed traits similar to Perciasacchi but a less uniform distribution of ash and minerals.

Considering the individual mineral decay during processing (Appendix A), the average losses from wholemeal through debranning to semolina for all genotypes over three environments differed among minerals evaluated in the present study, ranging from 19.7% for S to 76.3% for Mn. Our findings were in accord with Cubadda et al. [37], the low S decay, observed in semolina suggest its evenly presence throughout the kernel as protein-bound (i.e., prolamins). A similar behavior was also observed for Mo and Na showing a decay in semolina of 28.7% and 24%, respectively, as also evidenced by De Brier et al. [38].

On the contrary, major decays were observed for Mn, Zn, Mg, Fe, and K rapidly lost during debranning treatments that exceed 60% in the semolina followed by Ca and Cu, with a decay of 53% and 46%, respectively. These results confirmed previously studies carried out by Ficco et al. [21], Cubadda et al. [37] and De Brier et al. [38].

Moreover, De Brier et al. [38] reported that the distribution of minerals depended on their root uptake, on growing conditions (rainfall, soil moisture and temperature) and on redistribution to the grain via the phloem. This different mineral localization determined that not all regions resulted easily accessible during debranning, explaining the differences observed in their accumulation after processing.

### 3.3. Relationship among Kernel Traits, Ash, and Mineral Content

To understand the relationship between seed morphological and yield traits, and ash content during debranning a Pearson’s correlation coefficient was determined (Table 2). No significant correlation was highlighted between the GY and the yield components or other grain characteristics (seed morphology and ash), indicating that this important agronomic trait (GY) mainly depends on the kernel number per unit area rather than the weight of the seeds [39].

Thickness and seed area were positively correlated with TKW (0.87 and 0.85 *p* < 0.01, respectively) and seed thickness was positively correlated with length (*r* = 0.74 *p* < 0.01). TKW and thickness showed both a significant negative correlation at *p* < 0.01 with ash during debranning (DK1–6) with values ranging from −0.86 to −0.89 (TKW vs. DK1–6) and from −0.74 to −0.84 (Thickness vs. DK1–6), respectively. Other authors previously evidenced in bread wheat a negative correlation of ash flour content with TKW (*r* = –0.85) [24] indicating high ash for small kernels are expected to have minor endosperm to bran ratio.

The correlations among the investigated parameters (morphological, yield and yield-related traits, ash, and sum of macro- and micro-elements), genotypes, environments, and treatments were studied by means of PCA. Figure 2 reports the loading plot and the biplots of the PCA models calculated on the average value of the investigated parameters for each genotype, environment, and treatment. The first two factors explained 65% of the total variability (37% and 28% for Factor 1 and Factor 2, respectively) (Figure 2a). The first factor was highly and positively associated with thickness (0.96), length (0.85), and TKW (0.81) while the second factor showed a positive association with macro- (0.95) and micro-elements (0.88), and ash content (0.95).

A separation between genotypes and environments was observed along the Factor 1 (Figure 2b). The PCA showed as Perciasacchi and Cappelli resulted located in the right of the graph and associated with high thickness, length, and TKW. Instead, Timilia and Russello resulted concentrated in the left of PCA and well separated in the three environments among each other. Regarding the debranning process, all genotypes showed for ash and minerals a directional gradient from wholemeal to semolina along Factor 2. The same behavior was observed previously by Ficco et al. [21], confirming the effectiveness of debranning and milling process in cereals to study ash content as well as macro- and micro-element concentration in wheat grain. These findings highlighted the importance of seed thickness on the distribution of ash content in durum wheat grain. Setting a limit of ash, Perciasacchi with the heavier and thicker seeds, followed by Cappelli, showed a substantial reduction in ash content in the first two debranning treatments (Figure 1) that could translate into a greater semolina yield.

## 4. Conclusions

These data introduced a novelty related to thickness; in fact, up to now, the positive correlation between kernel size and kernel weight and, indirectly, TKW, allowed to select on the basis of large and spherical kernels, having greater endosperm to bran ratio, positively affecting yield and milling performance. Since thickness was strongly affected by the genotype, it could represent a simple trait that can be used in the early generations of breeding programs to predict the milling yield.

Theoretical models to predict semolina yield could be increased by optimizing grain size and shape with large and thickness grains being the optimum grain morphology.

## Figures and Tables

**Figure 1 foods-09-01523-f001:**
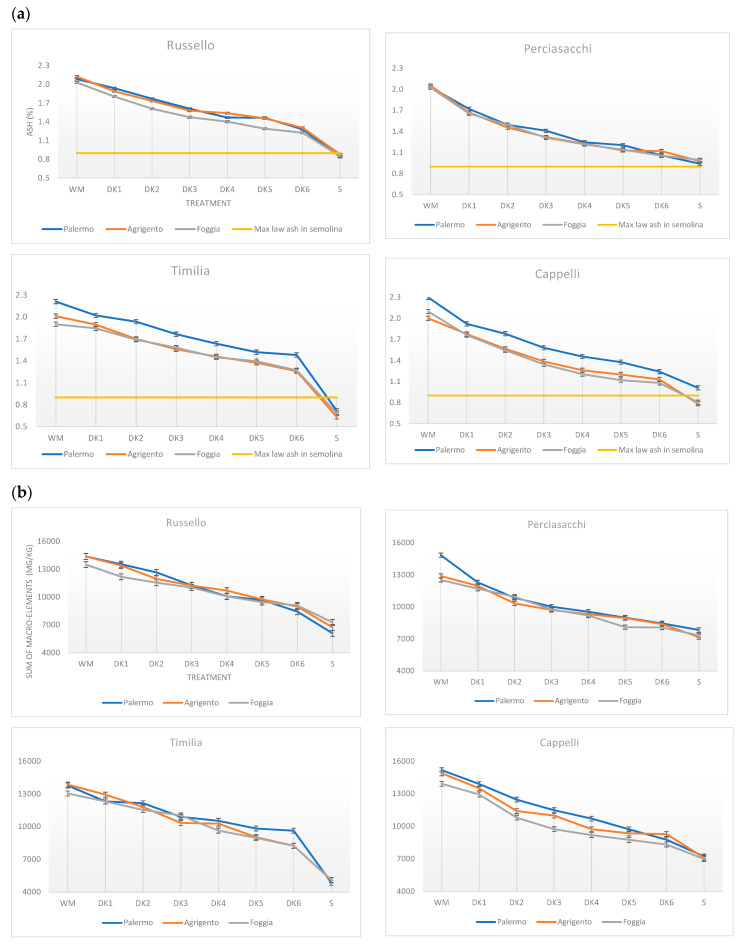
Effect of processing (debranning and milling) for four old durum wheat genotypes over three environments: (**a**) ash, (**b**) sum of macro- and (**c**) sum of micro-elements. WM, wholemeal; S, semolina; DK-1 to DK-6, debranned kernels. The bars represent the standard error of the E × T. The red dashed line represents the maximum level of ash equal to 0.9% required by Italian law (DPR 187/2001). (For interpretation of the references to color in this figure legend, the reader is referred to the web version of this article).

**Figure 2 foods-09-01523-f002:**
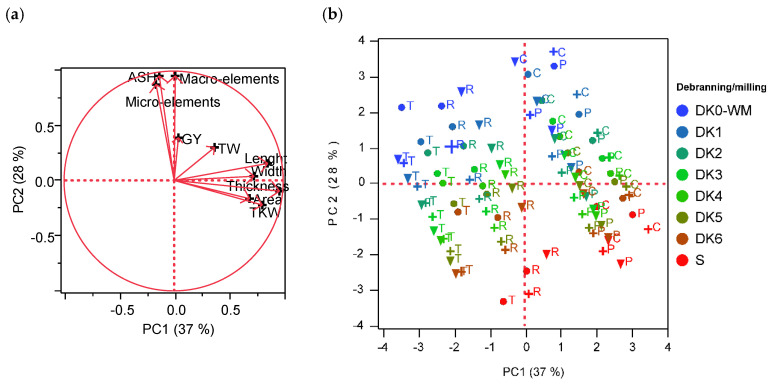
PCA loading plot showing the distribution of the analyzed variables (**a**) and score plot (**b**) showing the distribution of old genotypes (C = Cappelli, P = Perciasacchi, R = Russello, T = Timilia), grown in the three environments (● Palermo, **+** Foggia, ▼ Agrigento) during processing (milling/debranning from blue to red: DK0-WM (wholemeal), DK1, DK2, DK3, DK4, DK5, DK6, S (semolina)) along the two principal factors. (For interpretation of the references to color in this figure legend, the reader is referred to the web version of this article). Macro-elements = sum of Na, K, P, Ca, and Mg; micro-elements = sum of Mn, Fe, Cu, Zn and Mo; GY = grain yield; TW = Test Weight; TKW = Thousand kernel weight; PCA = Principal component analysis.

**Table 1 foods-09-01523-t001:** Morphological by Image analysis and yield traits on durum wheat grains, and ash on wholemeal recorded in three environments.

Genotype (G)	Environment (E)	Length (mm)		Width (mm)		Thickness (mm)		Area (mm^2^)		TKW (g)		TW (kg/hL)		GY (t/ha)		Ash (%)	
Cappelli	Palermo	7.55 ± 0.01		3.36 ± 0.01		3.05 ± 0.04		16.46 ± 0.03		48.23 ± 0.01		80.50 ± 1.27		3.63 ± 0.13		2.30 ± 0.04	
	Agrigento	7.32 ± 0.01		3.37 ± 0.01		3.09 ± 0.01		16.44 ± 0.01		50.33 ± 0.01		80.25 ± 1.27		3.32 ± 0.66		2.00 ± 0.10	
	Foggia	7.33 ± 0.01		3.49 ± 0.02		3.16 ± 0.01		17.35 ± 0.01		54.00 ± 0.28		79.40 ± 0.78		2.98 ± 0.27		2.10 ± 0.01	
Mean Cappelli		7.40		3.41		3.10		16.75		50.85		80.05		3.31		2.13	
Russello	Palermo	7.01 ± 0.06		3.20 ± 0.01		2.94 ± 0.01		16.24 ± 1.28		40.63 ± 0.04		81.16 ± 1.05		3.08 ± 0.25		2.09 ± 0.01	
	Agrigento	7.19 ± 0.03		3.24 ± 0.02		2.99 ± 0.01		15.96 ± 0.71		42.44 ± 0.62		80.73 ± 1.31		3.21 ± 0.60		2.13 ± 0.05	
	Foggia	7.04 ± 0.09		3.22 ± 0.04		2.96 ± 0.01		16.19 ± 1.34		43.69 ± 0.26		76.98 ± 0.39		2.78 ± 0.09		2.03 ± 0.01	
Mean Russello		7.08		3.22		2.96		16.13		42.25		79.62		3.02		2.08	
Timilia	Palermo	6.88 ± 0.11		3.14 ± 0.06		2.90 ± 0.01		16.54 ± 0.30		37.48 ± 0.04		79.85 ± 0.92		3.42 ± 0.08		2.21 ± 0.03	
	Agrigento	6.79 ± 0.01		3.20 ± 0.01		2.91 ± 0.01		14.26 ± 0.01		39.97 ± 0.03		76.95 ± 1.27		2.98 ± 0.08		2.01 ± 0.06	
	Foggia	6.69 ± 0.01		3.18 ± 0.01		2.85 ± 0.01		16.54 ± 0.71		40.46 ± 0.06		77.45 ± 0.21		2.72 ± 0.24		1.90 ± 0.14	
Mean Timilia		6.79		3.17		2.89		15.78		39.30		78.08		3.04		2.04	
Perciasacchi	Palermo	7.47 ± 0.06		3.23 ± 0.01		3.12 ± 0.03		19.27 ± 0.54		62.55 ± 0.06		78.75 ± 0.42		3.15 ± 0.57		2.03 ± 0.01	
	Agrigento	7.03 ± 0.01		3.27 ± 0.03		3.10 ± 0.01		19.33 ± 0.18		64.38 ± 0.18		77.93 ± 0.25		2.27 ± 0.39		2.05 ± 0.01	
	Foggia	7.04 ± 0.04		3.21 ± 0.01		3.12 ± 0.01		17.74 ± 0.14		63.07 ± 0.04		79.53 ± 0.67		2.53 ± 0.21		2.03 ± 0.04	
Mean Perciasacchi		7.18		3.24		3.11		18.78		63.33		78.74		2.65		2.04	
	Mean Palermo	7.22		3.23		3.00		17.13		47.22		80.07		3.32		2.16	
	Mean Agrigento	7.08		3.27		3.02		16.50		49.28		78.97		2.95		2.05	
	Mean Foggia	7.02		3.28		3.02		16.96		50.31		78.34		2.75		2.02	
Analysis of variance		F	*p*	F	*p*	F	*p*	F	*p*	F	*p*	F	*p*	F	*p*	F	*p*
G		163.3	***	89.9	***	342.2	***	26.5	***	14343.9	***	5.6	*	3.4	ns	3.5	*
E		36.4	***	6.0	*	5.9	*	2.1	ns	406.2	***	7.3	**	5.2	*	13.5	***
G × E		15.1	***	5.0	*	14.0	***	3.9	*	61.4	***	4.5	*	0.8	ns	5.6	**

Data are represented as mean ±standard deviation. G, E and G × E are measured at *p* < 1%; ns, not significant; *, ** and *** represents significance at *p* < 0.05, *p* < 0.01 and *p* < 0.001, respectively. TKW: Thousand kernel weight; TW: Test weight; GY: Grain yield.

**Table 2 foods-09-01523-t002:** Correlations among morphological and productive traits and ash.

	GY	TW	TKW	Length	Width	Thickness	Area	Ash-WM	Ash-DK1	Ash-DK2	Ash-DK3	Ash-DK4	Ash-DK5	Ash-DK6	Ash-S
GY	1.00														
TW	0.58	1.00													
TKW	−0.45	−0.04	1.00												
Length	0.47	0.49	0.47	1.00											
Width	0.18	0.25	0.36	0.67	1.00										
Thickness	−0.13	0.24	**0.87**	**0.74**	0.69	1.00									
Area	−0.39	0.01	**0.85**	0.37	0.17	0.66	1.00								
Ash-WM	0.64	0.58	−0.14	0.47	0.21	0.1	−0.06	1.00							
Ash-DK1	0.68	0.3	**−0.89**	−0.23	−0.31	**−0.74**	−0.69	0.5	1.00						
Ash-DK2	0.64	0.33	**−0.86**	−0.27	−0.38	**−0.74**	−0.63	0.54	**0.98**	1.00					
Ash-DK3	0.57	0.19	**−0.86**	−0.35	−0.52	**−0.83**	−0.58	0.41	**0.96**	**0.97**	1.00				
Ash-DK4	0.49	0.16	**−0.88**	−0.40	−0.55	**−0.84**	−0.65	0.4	**0.93**	**0.95**	**0.97**	1.00			
Ash-DK5	0.52	0.24	**−0.87**	−0.37	−0.56	**−0.85**	−0.63	0.34	**0.94**	**0.94**	**0.97**	**0.98**	1.00		
Ash-DK6	0.43	0.13	**−0.86**	−0.47	−0.53	**−0.84**	−0.59	0.38	**0.92**	**0.94**	**0.95**	**0.97**	**0.93**	1.00	
Ash-S	−0.12	0.34	0.71	0.64	0.25	0.67	0.64	0.37	−0.49	−0.42	−0.47	−0.44	−0.43	−0.50	1.00

Bold correlations are significant at *p* < 0.01. WM, wholemeal; DK-1 to DK-6, debranned kernels; S, semolina. GY: Grain yield; TW: Test weight; TKW: Thousand kernel weight.

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
