# Peer review of "Relationship between Seed Morphological Traits and Ash and Mineral Distribution along the Kernel Using Debranning in Durum Wheats from Different Geographic Sites"

_foods, 2020, doi:10.3390/foods9111523_

Round 1

Reviewer 1 Report

The grain debranning  is useful treatment for flour production of improved pro-health quality. Earlier research of the Authors (Ficco et al. 2020) showed that removing the external layers of grain makes the flours rich in bioactive compounds and low in toxic ones. The debranning treatment performed under strictly defined laboratory conditions can also be a useful research tool to study the distribution of these compounds throughout anatomical layers of  the kernel and the latter is the subject of the reviewed manuscript.

Within four durum wheat cultivars grown in three regions of  Italy and using six debranning levels, Authors examined the relationship between the morphological features of the kernel and distribution of the total ash and macro- and micro-elements concentration in kernel layers (from pericarp to endosperm). Also application aspect of research, i.e. selection of durum wheat genotypes with more favorable ash distribution in the kernel could increase semolina yield, can be fully appreciated.

The shortcomings of this manuscript include the failure to perform calculations for the twelve experimental combinations (4 cultivars x 3 growing regions) of grain weight losses depending on the debranning time (using non-linear regression models) and not taking into account these data when correcting the ash content. This is important because the experimental combinations cause significant differences in the morphology and anatomy of the kernel, and in particular in the texture (hardness) of the endosperm. In turn, all these features contribute to a significant differentiation in the kernel weight loss during debranning. This  thesis is confirmed when we confront the data published by Ficco et al. 2020 (5.4, 10.4, 13.7, 17.0, 19.4 and 23.5 %) with the ones currently being reviewed (5.0, 10.4, 15.7, 19.7, 23.1 and 26.5 %). Thus, a more objective way of illustrating the distribution of macro- and micro-elements along the kernel would be graphs of the dependence of ash content on the kernel weight loss.

The detailed remarks:

Title of the manuscript is inappropriate (not effects but relationships between these features have been examined, this correction applies throughout the text of the manuscript) and does not include the specificity (debraning treatment as a tool for determining the ash distribution along the kernel) and the scope of the research carried out (effects of wheat genotype and growing regions).

Line13-14, rewrite sentence: ..to study the ash distribution … .

Line 101-102,  despite the fact that "semolina yield" is the most important indicator in the evaluation of the  milling value of  durum wheat, Authors did not determine its value for the tested wheat samples, why?

Line 168-170, “Figure 1 shows …”.

Line 182-184, “ … and semolina production, in relation to DK6, …”.

Fig. 1c, graphs for the varieties Timilla and Cappelli are not shown.

Author Response

Response to Reviewer 1 Comments

Reviewer 1

The shortcomings of this manuscript include the failure to perform calculations for the twelve experimental combinations (4 cultivars x 3 growing regions) of grain weight losses depending on the debranning time (using non-linear regression models) and not taking into account these data when correcting the ash content. This is important because the experimental combinations cause significant differences in the morphology and anatomy of the kernel, and in particular in the texture (hardness) of the endosperm. In turn, all these features contribute to a significant differentiation in the kernel weight loss during debranning. This  thesis is confirmed when we confront the data published by Ficco et al. 2020 (5.4, 10.4, 13.7, 17.0, 19.4 and 23.5 %) with the ones currently being reviewed (5.0, 10.4, 15.7, 19.7, 23.1 and 26.5 %). Thus, a more objective way of illustrating the distribution of macro- and micro-elements along the kernel would be graphs of the dependence of ash content on the kernel weight loss.

Our reply: We thank the Reviewer 1 for the exhaustive overview of the manuscript and suggestions.  

We agree that the grain weight loss depends on the debranning time, genotype and environment. So we have tried to clarify, in accordance also to the Reviewer 2, the paragraph 2.2 of materials and methods. For each genotype, environment, and time of debranning we have measured the weight of each debranned grain and of the removed material, measuring the percentage respect to initial weight. The mean of all data represents the 5.0, 10.4, 15.7, 19.7, 23.1 and 26.5 % of removed bran layer used for the Fig. 1 which have incorporated the differences in morphological parameters. In this direction, we have improved the results and discussion section.

Title of the manuscript is inappropriate (not effects but relationships between these features have been examined, this correction applies throughout the text of the manuscript) and does not include the specificity (debraning treatment as a tool for determining the ash distribution along the kernel) and the scope of the research carried out (effects of wheat genotype and growing regions).

Our reply: We agree with Reviewer 1 comments. In fact, we have re-visited the title “Effect of kernel shape and size on ash and mineral content in durum wheat”. So, the new title was “Relationship between seed morphological traits and ash and mineral distribution along the kernel using debranning in durum wheats from different geographic sites”. We have applied the substitution of the ‘effects’ with ‘relationship’ in all the text.

Line13-14, rewrite sentence: ..to study the ash distribution … .

Our reply: The sentence has been modified as “Debranning was applied to durum wheat to the study the relationship between kernel shape and size and ash and mineral distribution having implications for semolina yield”. (Now lines 18-19).

Line 101-102, despite the fact that "semolina yield" is the most important indicator in the evaluation of the milling value of durum wheat, Authors did not determine its value for the tested wheat samples, why?

Our reply: We have no reported semolina yield as it is an indirect trait strictly correlated to ash on which we have concentrated our determinations.

Line 168-170, “Figure 1 shows …”.

Our reply: We have changed “showed” by “shows”, as requested. (Now line 197).

Line 182-184, “ … and semolina production, in relation to DK6, …”.

Our reply: We have added “in relation to DK6”. The new sentence is “The decrease was greater after the first debranning treatment (DK1) respect to wholemeal and at semolina production, in relation to DK6, with some exceptions”. (Now lines 248-249).

Fig. 1c, graphs for the varieties Timilla and Cappelli are not shown.

Our reply: The Fig. 1 was completed with Timilia and Cappelli graphs.

Reviewer 2 Report

Major remarks

The introduction must be improved in sofar as it is difficult for the reader to clearly identify the innovative character of the approach envisaged. Likewise, the general discussion of the results obtained must be enriched in order to really bring out the new scientific elements resulting from the work. Can we reasonably extrapolate the results obtained from the 5 genotypes selected to set up a global breeding program?

General remarks

A certain number of editorial requirements do not seem to be respected by the authors in this version of the text: referencing of bibliographic elements in the text, description of additional material, author contributions ...

Minor remarks

Paragraph 2.1 line 74: Why did the authors choose to use four old tetraploid durum wheat for this study? What are the scientific arguments ?

Paragraph 2.2 line 98: On what basis of calculation are the quoted debranning percentages obtained?

Paragraph 3.1 line 139: Can the authors clarify the following sentence: "old varieties characterized by a late cycle of development"?

Author Response

Response to Reviewer 2 Comments

Reviewer 2

Major remarks

The introduction must be improved in sofar as it is difficult for the reader to clearly identify the innovative character of the approach envisaged. Likewise, the general discussion of the results obtained must be enriched in order to really bring out the new scientific elements resulting from the work. Can we reasonably extrapolate the results obtained from the 5 genotypes selected to set up a global breeding program?

Our reply: We thank the Reviewer 2 for his/her considerations. The introduction and discussion sections have been improved, adding three references (Baasandorj et al., 2015; Posner et al., 2005; Shetlar et al., 1947).

The choice of a limited number of varieties with contrasting seed characteristics has been linked to the need to take into account the genetic variability of the species, but it have given us the possibility of evaluating the interaction with the environment, which is much more difficult to achieve with a large number of genotypes. A deeper study will extend to a large number of genotypes, representative of the durum wheat germplasm. 

General remarks

A certain number of editorial requirements do not seem to be respected by the authors in this version of the text: referencing of bibliographic elements in the text, description of additional material, author contributions ...

Our reply: We thank the Reviewer for the general remarks. We have followed the authors guidelines for the references in the text and the supplementary material while author contributions were inserted during the submission of the manuscript.

Minor remarks

Paragraph 2.1 line 74: Why did the authors choose to use four old tetraploid durum wheat for this study? What are the scientific arguments?

Our reply: We have chosen to study the old durum wheat genotypes as they are all characterized by stable grain yield and by different morphological traits. Furthermore, there is a consolidated evidence of a growing appreciation of old durum wheat genotypes among niche consumers, thanks to nutritional and health properties and to the low-input agricultural systems typical of marginal areas.

Paragraph 2.2 line 98: On what basis of calculation are the quoted debranning percentages obtained?

Our reply: The debranning was monitored through time control. The treatments were non-debranned kernels and six debranning times (30 s, 60 s, 90 s, 120 s, 150 s, 180 s). Samples were weighted before and after debranning, measuring the weight of grains after each debranned time and the weight of removed materials. To measure the quote of debranning for each genotype, the weight of the debranned grain and of the removed material at each debranning time was weighted and their percentage respect to initial weight was calculated.

So, the paragraph 2.2 was improved adding the sentence “The weight of each debranned grain and the removed material at each debranning time was weighted and their percentage respect to initial weight was calculated”. (Now lines 123-124).

Paragraph 3.1 line 139: Can the authors clarify the following sentence: "old varieties characterized by a late cycle of development"? 

Our reply: Old varieties were tall, prone to lodging, and late in flowering. Instead, the main strategy for modern plant breeding programs aimed at better grain yield, shorter stature, early flowering to obtain resistance to lodging and early ripening in order to tackle better the drought season.

Therefore we have modified the text as “old varieties characterized by a long vegetative phase and short grain filling phases, reflecting the higher plant height above 100 cm and the low grain yield, as were all the varieties considered in present study”. (Now lines 169-170).

Round 2

Reviewer 2 Report

No additional comments